# In Vitro Determination of the Skin Anti-Aging Potential of Four-Component Plant-Based Ingredient

**DOI:** 10.3390/molecules27228101

**Published:** 2022-11-21

**Authors:** José Quiles, Maria Cabrera, Jonathan Jones, Menelaos Tsapekos, Nuria Caturla

**Affiliations:** 1Research Projects Department, Life Length SL, Calle Miguel Angel 11, 28010 Madrid, Spain; 2Research and Development Department, Monteloeder SL, Miguel Servet 16, 03203 Elche, Spain

**Keywords:** plant extracts, antiaging, telomere length, antioxidant agents, telomerase, antiglycation, cell proliferation, Herba Cistanche, *Centella asiatica*, pomegranate, sweet orange

## Abstract

The beauty industry is actively searching for solutions to prevent skin aging. Some of the crucial elements protecting cells from the aging process are telomere shortening, telomerase expression, cell senescence, and homeostasis of the redox system. Modification of these factors using natural antioxidants is an appealing way to support healthy skin aging. Therefore, in this study, we sought to investigate the antiaging efficacy of a specific combination of four botanical extracts (pomegranate, sweet orange, Cistanche and *Centella asiatica*) with proven antioxidant properties. To this end, normal human dermal fibroblasts were used as a cell model and the following studies were performed: cell proliferation was established by means of the MTT assay and the intracellular ROS levels in stress-induced premature senescence fibroblasts; telomere length measurement was performed under standard cell culture conditions using qPCR and under oxidative stress conditions using a variation of the Q-FISH technique; telomerase activity was examined by means of Q-TRAP; and AGE quantification was completed by means of ELISA assay in UV-irradiated fibroblasts. As a result, the botanical blend significantly reversed the H_2_O_2_-induced decrease in cell viability and reduced H_2_O_2_-induced ROS. Additionally, the presence of the botanical ingredient reduced the telomere shortening rate in both stressed and non-stressed replicating fibroblasts, and under oxidative stress conditions, the fibroblasts presented a higher median and 20th percentile telomere length, as well as a lower percentage of short telomeres (<3 Kbp) compared with untreated fibroblasts. Furthermore, the ingredient transiently increased relative telomerase activity after 24 h and prevented the accumulation of UVR-induced glycated species. The results support the potential use of this four-component plant-based ingredient as an antiaging agent.

## 1. Introduction

The world’s population is aging at a rapid rate, which has important consequences in healthcare. Consequently, this is gaining significant attention, and science continues to uncover effective approaches to support healthy aging and maintain quality of life for as long as possible.

The skin is the largest and one of the most important organs in the body, protecting us from external elements and potential harm, helping to regulate body temperature, and allowing us to interact with the environment through the sense of touch. Skin aging has gained interest not only because it is the most visual manifestation of the aging process, but also because it represents a picture of overall human health.

Skin aging is a complex biological process influenced by genetic determinants (intrinsic aging) and chronic exposure to several environmental factors, such as UV radiation and mechanical stress (extrinsic aging)., etc. The skin undergoes aging as a gradual process associated with the loss of certain fundamental skin properties that eventually result in wrinkles, laxity, and changes in pigmentation, among others [1]. At the structural level, the most relevant changes associated with aging are increased collagen degradation and reduced collagen biosynthesis, which in turn result in a net collagen deficiency, as well as aberrant production of other scaffolding molecules, such as elastin, proteoglycans and glycosaminoglycans [2].

At the molecular level, skin aging is a complex and multifactorial process mainly caused by an imbalanced redox status, DNA mutation, mitochondrial dysfunction, telomere shortening, cell senescence, and the deposition of advanced glycation end products (AGEs) within the dermis, among other things [3].

Aging was first described by Hayflick and Moorhead, who demonstrated that somatic mammalian cells have a limited ability to divide [4]. Consequently, cells lose their proliferative capacity and enter a state of irreversible cell cycle arrest, later termed replicative senescence [5,6]. As we age, senescent skin fibroblasts tend to prevail, which are characterized by their inability to proliferate. Furthermore, in senescence they begin to secrete proinflammatory cytokines, catabolic modulators such as matrix metalloproteinases (MMPs), and reactive oxygen species, resulting in a decline in the function and appearance of skin, as well as a decrease in the skin’s natural self-regenerative potential, along with reduced collagen and elastin fibers [7,8].

The molecular mechanism behind replicative senescence is thought to be at least partially associated with telomere shortening [9,10]. Telomeres are repetitive DNA sequences (TTAGGG) that protect chromosomes during cell division and are essential for DNA replication. However, telomeres shorten in each replication process, and once they reach a certain length, the cells no longer divide and enter replicative senescence. This issue can be compensated either by directly protecting the telomere and minimizing its attrition or by the enzyme telomerase, which can slow or even reverse telomere shortening. Telomerase activators enhance the efficiency of the DNA repair process and protect cells from stress and DNA-damaging conditions [11]. Telomere length predicts the replicative capacity of human fibroblasts [12], and the skin is particularly susceptible to accelerated shortening because of both proliferation and external DNA-damaging agents such as exposure to solar radiation, pollution or reactive oxygen species (ROS) [13,14].

Dermal fibroblasts play a key role in maintaining skin homoeostasis by synthesizing and degrading extracellular matrix components. Fibroblasts are affected in different ways throughout aging; for instance, an important hallmark of chronological skin aging is a lower proliferation rate [15]. Additionally, aged fibroblasts are characterized by the reduced expression of genes encoding collagen isoforms, as well as increased sensitivity to oxidative stress and higher β-galactosidase activity, among other features [16,17,18]. Therefore, in vitro cultured fibroblasts represent a valuable model to evaluate the impact of chronological aging and the throughput screening of putative anti-aging compounds [18].

The use of herbs, botanicals and their constituents as nutraceuticals and cosmeceuticals have gained credibility and popularity, leading to the establishment of the aging-related nutraceutical market, which is one of the largest consumer markets today [19]. In recent years, numerous in vitro and in vivo studies, as well as clinical trial interventions, have displayed the beneficial effects of plant phenolic compounds to slow down or even prevent aging-associated deterioration of skin appearance and function [20,21,22]. Beyond their highly studied antioxidant and anti-inflammatory effects, recent research has highlighted the modulatory capacity of plant-derived compounds on several cellular pathways of interest, and, of note, on regulating cellular senescence, telomere shortening and AGE accumulation [7,23,24,25].

Different scientifically published articles have proven the antioxidant and anti-inflammatory properties of the phenolic compounds present in *Centella asiatica* [26,27], *Punica granatum* fruit [28,29], Herba Cistanche [30], and sweet orange extract [31], which may benefit skin due to their ability to modulate or reduce different cellular pathways involved in the skin aging process. We hypothesize that such properties attributed to these botanical sources and their polyphenolic compounds could be beneficial for reversing important cellular and molecular changes suffered by skin cells during aging, such as increased senescence due to telomere shortening and increased ROS and AGE accumulation. In addition, the potential beneficial effects on the skin health of these four botanical sources have also been reported. For instance, the hesperidin present in sweet orange has multiple potential benefits for cutaneous functions, including wound healing, UV protection, antimicrobial, and skin lightening [32]. *C. asiatica* extract and its triterpenoids have been extensively used in dermo-cosmetics to treat and relieve acne, wounds and inflammatory skin conditions such as psoriasis and atopic dermatitis [26,33]. Regarding *P. granatum*, in vitro and in vivo studies have demonstrated that the topical application and oral consumption of pomegranate fruit could reduce facial photodamage and ultraviolet-induced pigmentation [28,29,34]. Finally, Herba Cistanche, a traditional Chinese medicine herb, has been widely used in East Asian countries to treat irradiation-related skin disorders [30].

Thus, in the present study, a specific combination of a pomegranate extract, enriched in punicalagins, sweet orange extract containing hesperidin, Herba Cistanche extract enriched in phenylpropanoids and *Centella asiatica* extract standardized in triterpenoids was used for the first time to study its antiaging efficacy on normal human dermal fibroblasts. To achieve these objectives, the capacity of this combination to prevent the loss of proliferative potential, ROS formation and telomere shortening on aged fibroblasts was studied. Additionally, we studied whether the above-mentioned blend was capable of reducing the presence of AGEs, an important process responsible for skin aging [35], as well as the potential anti-melanogenic effects on human melanocytes exposed to UVA.

## 2. Materials and Methods

### 2.1. Reagents

Normal human epidermal melanocyte growth medium, low-glucose Dulbecco’s modified Eagle’s medium (DMEM), fetal bovine serum (FBS), and trypsin/EDTA were purchased from Gibco. Primary fibroblast-specific growth medium and supplements were obtained from Promocell. Trypan blue solution and the Protein Assay Kit II were purchased from Bio-Rad. Phosphate buffered saline, ethanol, DMSO, distilled water, hydrogen peroxide (H_2_O_2_), trypsin, NaOH, NaCl, sodium deoxycholate, sodium dodecyl Sulfate (SDS), L-glutamine, the Fluorometric Intracellular ROS Kit and CHAPS (3-cholamidopropyl dimethylammonium 1-propanesulfonate) lysis buffer were purchased from Sigma-Aldrich. DNAse-I and RNeasy extraction kits were purchased from Qiagen. The PrimeScript™ RT Reagent Kit (Perfect Real Time) was purchased from Takara Clontech. MTT powder and PureLink Genomic DNA Mini Kit were obtained from Invitrogen. Bovine serum albumin (BSA), Triton X-100 and Tris base were purchased from Fisher Scientific. SYBR^®^ Green qPCR Master Mix were obtained from Applied Biosystems. The Pierce^®^ BCA Protein Assay Kit and protease inhibitor tablets were obtained from Thermo Scientific. The Absolute Human Telomere Length Quantification qPCR Assay Kit was purchased from ScienCell’s. The OxiSelect™ Advanced Glycation End Product (AGE) ELISA Kit was purchased from Cell Biolabs. FastStart essential DNA green master was purchased from Roche Life Science. Commercial primary reference HPLC standards, asiaticoside and verbascoside, were obtained from The United States Pharmacopeial Convention (USP), equinacoside was purchased from Sigma-Aldrich, punicalagin A+B was purchased from PhytoLab GmbH & Co and hesperedin Ph. Eur. was obtained from the Council of Europe.

### 2.2. Experimental Product

The assessed product was a registered optimized blend of 4 polyphenolic extracts from *Centella asiatica,* pomegranate (*Punica granatum*) fruit, sweet orange (*Citrus aurantium var. sinensis*) and Herba Cistanche stem (called Eternalyoung^®^, EY, Monteloeder SL, Elche, Spain). In total, *w*/*w*, this blend comprised a minimum content of: 2.5% verbascoside; 12.5% hesperidin; 3% punicalagins and 7% asiaticosides. Table 1 and Figure 1 shows representative chromatograms of the ingredients with the major compounds identified.

A Stock solution of 10 mg/mL of the tested sample was prepared in DMSO and sequentially diluted in culture medium to add to the cells at the desired concentration in each of the experiments (the final concentration used is detailed in its corresponding method description.). Solutions were sterilely filtered and freshly prepared for every cellular assay. The final concentration of DMSO in the cell culture medium was a maximum of 0.1%, at which no detrimental effect on cell growth or toxicity was detected (data not included).

### 2.3. Cell Culture

Normal human dermal fibroblasts (NHDFs) were cultured in PromoCell fibroblast growth medium supplemented with L-penicillin-streptomycin and 10% fetal bovine serum (Gibco). The medium was renewed every 2–3 days and cells were passaged when fibroblasts reached 70–80% confluency.

Normal human epidermal melanocytes (NHEMs) were cultured in normal human melanocyte growth medium with supplements (Gibco). The medium was renewed every 2–3 days and cells passaged when melanocytes reached 70% confluency.

Both NHDF and NHEM cultures were maintained at 37 °C with a 5% CO_2_ humidified atmosphere.

### 2.4. High-Performance Liquid Chromatography (HPLC) Analysis of EY

The composition of the combination of extracts in EY was identified and quantified using an HPLC instrument (Agilent 1260 Infinity; Agilent Technologies, Inc., Palo Alto, CA, USA) coupled with a photodiode array detector. A BDS Hypersil C18 (5 μm, 250 × 4 mm) column (Thermo Fisher, Waltham, MA, USA) was used for analytical purposes. The temperature was set at 25 °C and the injection volume was 20 µL. The compounds were identified by comparing the retention times and ultraviolet (UV) spectra of the peaks of the HPLC/PDA chromatograms to those of commercially available standards (Std). The quantification was performed according to the following formula: Assay sample (%) = (Area Sample/Area Std) × (Conc. Std/Conc. Sample) × Assay Std.

Three different HPLC methods were used to ensure the correct identification and quantification of different compounds present in the final blend: Method 1 for phenylpropanoid and the citrus flavone where the area of the peaks was determined at a wavelength of 330 nm for phenylpropanoids (verbascoside and echinacoside) and 280 nm for the citrus flavone hesperidin; Method 2 for the identification and quantification of total punicalagins as a sum of punicalagin A and punicalagin B; and Method 3 for the identification and quantification of the glycosylated triterpene of *Centella asiatica* (asiaticosides). Total asiaticosides was calculated as the sum of asiaticoside, asiaticoside B and madecassoside using asiaticoside as standard. More details of the methods can be found in the Appendix A.

### 2.5. Human Fibroblast Proliferation by MTT Assay in Aging Conditioned Medium

The MTT assay allows the simple, accurate and reliable counting of metabolically active cells [36,37,38]. Since mitochondrial activity is directly related to the number of viable cells, this assay is usually performed to assess the proliferation capacity and cytotoxic potential of different compounds in different cell lines [38]. The MTT assay reflects the effects produced by a substance or treatment upon cell viability, which may be interpreted as toxic effects (cytotoxicity) if cell viability is compromised or stimulating effects (proliferation) if cell viability increases, comparing the treatments with the untreated control group.

To perform cell counting of live cells, cell viability was checked by staining with trypan blue solution. Live cells were counted in a Bürker chamber under the microscope.

To emulate the aging process, normal human dermal fibroblasts (NHDFs) were cultured overnight at a density of 8000 cells/well in a 96 well plate in growth medium. Twenty-four hours later, conditioned medium containing 500 μM hydrogen peroxide (H_2_O_2_) was added to the cells for 3 h, according to previously published work showing senescence induction in that range of doses [39,40]. After the incubation period, the medium was replaced by fresh medium containing the ingredient at different concentrations (0.01%, 0.005%, 0.001% and 0.0005%) for 96 h. Epidermal growth factor (EGF, 20 ng/mL) was included as a positive control and NHDF without H_2_O_2_ pre-treatment as a non-aging control (Control). The medium in all conditions contained only 0.5% FBS. After 96 h of incubation, the medium was removed, wells were washed with PBS and cell viability was then quantified using the MTT ((3-(4,5-Dimethylthiazol-2-yl)-2,5-diphenyltetrazolium bromide) assay. The MTT assay was performed following the ECVAM Guidelines as established in the ECVAM Database Service on Alternative Methods to Animal Experimentation (MTT assay protocol nr. 17) [41]. In brief, MTT solution 1:11 was added to each well. Plates were incubated in the refrigerated incubator at 37 °C for 3 h. The MTT reactive was removed, and DMSO 100% was added to each well to solubilize formazan crystals prior to absorbance measurements at 550 nm and 620 nm as a reference on a scanning multi-well spectrophotometer. A biological replicate with 8 technical replicates per condition was performed. The data were statistically analyzed using an ordinary one-way ANOVA test and Student’s *t*-test. Statistical significance was set at *p* < 0.05, with a 95% confidence interval. Absorbance values lower than those of control cells indicated a reduction in the rate of cell proliferation. Conversely, a higher absorbance rate indicated an increase in cell proliferation.

### 2.6. Measurement of Intracellular ROS Formation in Aging Conditioned Medium

NHDF cells were cultured overnight at a density of 10,000 cells/well in a 96-well plate in growth medium. Twenty-four hours later, conditioned medium containing 500 μM hydrogen peroxide (H_2_O_2_) was added to cells for 3 h, according to a previous study, to induce senescence. After the incubation period, the medium was replaced with conditioned medium taken from EY cultured (0.01%, 0.005%, and 0.0005%) for 24 h. In parallel, untreated NHDFs (no H_2_O_2_ and no products) were cultured for the same period. ROS production was then measured using a Fluorometric Intracellular ROS Kit that was added to wells following the manufacturer’s instructions and incubated for 1 h at 37 °C. After incubation, fluorescence was measured at λex = 540 nm/λem = 570 nm. Fluorescence values lower than those of control cells indicated a reduction in the rate of ROS generation. Conversely, a higher fluorescence rate indicated an increase in ROS generation. A biological replicate with 8 technical replicates per condition was performed. All data were statistically analyzed using an ordinary one-way ANOVA test and Student’s *t*-test. Statistical significance was set at *p* < 0.05, with a 95% confidence interval.

### 2.7. Telomere Length Measurement in Human Fibroblast: Real-Time PCR

Prior to cell seeding, cell numbers and viability were determined using trypan blue staining and counting in a Bürker chamber under the microscope.

For telomere length assessment, NHDF cells were cultured at a density of 500,000 cells/well in T25 flasks. EY was added to flasks at 0.0001% or 0.0005% concentrations 24 h after each cell passage, and changed for fresh product every 3 days, until cells reached 80% confluence. At this point, 500,000 cells were passaged again to a T75 flask and treatments commenced 24 h after passage. This process was repeated until untreated control cells performed 24 cell doublings. Cell doublings or divisions were estimated in every passage using the expression: # doublings = log2 (N/500,000), where N is the cell number before every seeding stage. After the incubation period, cells were collected and gDNA was extracted using PureLink Genomic DNA Mini Kit. Telomere length was analyzed using the ScienCell’s Absolute Human Telomere Length Quantification qPCR Assay Kit (AHTLQ) as per the manufacturer’s instructions. Genomic DNA (5 ng) was amplified with the FastStart Essential DNA Green using a QuantStudio 5 Quantitative real-time PCR (Applied Biosystem). gDNA was also extracted and amplified from cells at the initial doubling.

AHTLQ is designed to directly measure the average telomere length of a human cell population. The telomere primer set recognizes and amplifies telomere sequences. The single copy reference (SCR) primer set recognizes and amplifies a 100 bp-long region on human chromosome 17 and serves as a reference for data normalization. The reference genomic DNA sample with known telomere length serves as a reference for calculating the telomere length of target samples. To perform raw data analysis, the Pfaffl method [42] was used to calculate the gene relative expression ratio to SCR (internal control-housekeeping gene). The mathematical model of the relative expression ratio in RT-PCR was calculated as follows: $ratio=\frac{\left(E_{\mathrm{target}}\right){\;}^{\Delta CPtarget\;\left(control-sample\right)}}{{\left(E_{\mathrm{ref}}\right)}^{\;\Delta CPref\;\left(control-sample\right)}},\;$where *E_target_* is the real-time PCR efficiency of target gene transcript; *E_ref_* is the real-time PCR efficiency of a reference gene transcript; Δ*CP_target_* is the difference between the CP deviation of control and the sample of the target gene transcript; and Δ*CP_ref_* is the difference between CP deviation and the sample of the reference gene transcript.

Three technical replicates were included in the assay. All of the data were statistically analyzed, comparing with the control at the beginning (not aged, just before treatment started, expressed as T = 0). All data were statistically analyzed using an ordinary one-way ANOVA test and Dunnett’s post hoc test. Statistical significance was set at *p* < 0.05, with a 95% confidence interval.

### 2.8. Telomere Length Measurement in Fibroblast Submitted to Oxidative Stress: TAT^®^ Technology

Telomere length was evaluated under oxidative cell culture conditions over an 8-week period in the presence of EY. For that purpose, adult human fibroblasts were cultured for 8 weeks with the tested sample at three different concentrations (0.001%, 0.0005%, 0.0001%). Cells were cultured and assayed under oxidative stress conditions adding 10 μM H_2_O_2_ as an agent of oxidative stress. Alterations in telomere lengths and telomere shortening rates (TSRs) were determined in each EY-treated group and compared to the control untreated group. Telomere length measurements were performed using Life Length’s proprietary Telomere Analysis Technology (TAT^®^). TAT^®^ measures telomere length using a high-throughput (HT) Q-FISH (quantitative fluorescent in situ hybridization) technique, as previously described [43,44]. This method is based on a quantitative fluorescence in situ hybridization method modified for measuring individual chromosomes of cells in interphase obtaining measurements from thousands of telomeres individually. Briefly, telomeres are hybridized with a fluorescent peptide nucleic acid probe (PNA) that recognizes three telomere repeats (sequence: Alexa 488-OO-CCCTAACCCTAACCCTAA, purchased from Panagene).

The images of the nuclei and telomeres are captured by a high-content screen system. The intensity of the fluorescent signal from the telomeric PNA probes that hybridize to a given telomere is proportional to the length of that telomere. The intensities of fluorescence are translated to base pairs (bp) through a standard regression curve which is generated using control cell lines with a known telomere length. TAT^®^ not only measures telomere length in absolute base pair units, but also provides an assessment of the whole distribution of telomere length and the percentage of short telomeres (<3 Kbp), allowing for a more comprehensive analysis of each sample. Thus, three distinct variables were determined during TAT^®^ analysis: median telomere length, 20th percentile telomere length, and the percentage of telomeres < 3 Kbp that can be perceived as the quantity of critically short telomeres that increase the risk of cells entering senescence. The telomere length distribution, median telomere length, 20th percentile and percentage of telomeres < 3 Kbp are calculated with Life Length´s proprietary program. All samples were run in quintuplicate, and statistical analysis of the data was performed using Student’s *t*-test. Statistical significance was set at *p* < 0.05, with a 95% confidence interval.

Additionally, a proliferation analysis of the cell culture was performed, and the aforementioned variables generated by TAT^®^ were later normalized according to the cell population doubling in order to determine the telomere shortening rate values for each cell group.

### 2.9. Relative Telomerase Activity

In this assay, telomerase activity is determined by quantitative polymerase chain reaction (PCR)-based telomeric repeat amplification protocol (Q-TRAP) in primary cultures of adult human fibroblast after 0, 6, 24 and 72 h and in the presence of the tested sample EY at the following non-toxic concentrations: 0.001%, 0.0005% and 0.0001%.

Q-TRAP is an accurate and sensitive PCR-based assay that enables the measurement of telomerase activity [45,46,47]. To perform the assay, cellular pellets from normal human fibroblast cells were lysed using the CHAPS lysis buffer for protein extraction [46]. Samples were stored at 4 °C and were used within 24 h. Protein quantification was performed in each sample using Bio-Rad protein assay Kit. A minimum protein concentration of 0.3 mg/mL was required to proceed with the analysis of the samples to guarantee consistent results. Telomerase protein extracts were then incubated at 27 °C for 30 min with a specific oligonucleotide substrate to allow the enzymatic addition of telomeric DNA repeats, TTAGGG, by endogenous telomerase.

Following the enzymatic reaction, telomerase extension products were then amplified and quantified by real-time qPCR using SYBR Green (a green fluorescent cyanine dye) [45,46,47]. PCR was initiated at 95 °C for 10 min, followed by a 40-cycle amplification (95 °C for 15 s, 60 °C for 60 s) and a melting curve stage step. Reactions were monitored and analyzed with QuantStudio 5 (Applied Biosystems, Foster City, CA, USA). Telomerase activity in cell lines or samples was calculated based on the threshold cycle (Ct). The telomerase-positive standard dilution series was plotted against the telomere protein concentration (r^2^ > 0.9) as a standard curve of Ct value. All samples were run in triplicate, and only those samples with sufficient protein were analyzed (>0.3 mg/mL).

The telomerase-positive standard dilution series was plotted against the telomere protein concentration (r^2^ > 0.9) as a standard curve of Ct values. The standard curve was generated by graphing threshold cycles (Ct values) of HeLa cell line standards against log of 1000, 333, 111, 37.03, 12.34, 4.11, 1.37, and 0.45 ng of protein (whole cell extract).

The results obtained from each treatment and at each time point were compared with the appropriate control group using Student’s *t*-test (independent samples *t*-test). Statistical significance was set at *p* < 0.05, with a 95% confidence interval.

### 2.10. Antiglycation Study in Human Fibroblasts. AGE Quantification-ELISA Assay

Prior to cell seeding, live cell count and cell viability were determined by staining with trypan blue solution. Live cells were counted in a Bürker chamber under the microscope.

NHDF cells were seeded in 6-well plates at a density of 300,000 cells/well and kept overnight at 37 °C. After 24 h, cells were treated with the tested ingredient at 0.01%, 0.005% and 0.001% concentrations for 24 h. Then, cells were exposed to UVB irradiation for 36 s (18 mJ/cm^2^). Non-irradiated controls were incubated at 37 °C during this time. Four hours after irradiation, samples were processed for advanced glycation end-product (AGE) quantification in the lysates. The AGE levels were quantified with a specific human ELISA kit following the manufacturer’s instructions. Four technical replicates were used per condition. ELISA standards were used to generate a standard curve and interpolate the sample values. In parallel, total protein abundance per condition was quantified using the bicinchoninic acid assay (BCA assay) to normalize the raw AGE levels obtained by ELISA. All data were statistically analyzed using an ordinary one-way ANOVA test and Dunnett’s post hoc test. Statistical significance was set at *p* < 0.05, with a 95% confidence interval.

### 2.11. Melanin Quantification in Human Melanocytes Irradiated with UVA

For melanin quantification, NHEM were cultured overnight at a density of 50,000 cells/well in a 24-well plate with growth media. Afterward, cells were cultured with the tested sample diluted at 0.0005% and 0.005% for 7 days. During the incubation period, cells were irradiated with UVA twice per day for 3 days (10 min; total dose = 5 J/cm^2^). After the last irradiation, cells were processed for melanin quantification through the addition of NaOH 1M and incubation for 1 h at 60 °C. Subsequently, absorbance measurement was performed at 405 nm using a spectrophotometric microplate reader, as previous authors have successfully used this technique for the same purpose [48,49]. For all conditions, 1 biological replicate with 5 technical replicates was used.

In parallel, cell viability was quantified through the MTT assay to normalize melanin levels with respect to the number of living cells to avoid false positives due to UVA-induced cytotoxicity and to evaluate the cell protection of the tested sample against UVA damage. One biological replicate with five technical replicates for all the conditions was used.

Data representation was performed using the irradiated control as a reference control to detect the efficacy of the compound reducing melanin synthesis. All data were statistically analyzed using ordinary one-way ANOVA and Dunnett’s post hoc test. Statistical significance was set at *p* < 0.05, with a 95% confidence interval.

## 3. Results

### 3.1. Characterization of EY by High-Performance Liquid Chromatography (HPLC)

As it can be seen in Figure 1 and Table 1, the use of three different methods allowed us to correctly and accurately identify the main compounds. The two major phenylpropanoids that were identified and quantified were verbascoside and equinacoside representing 9% *w*/*w*, with the verbascoside concentration being 2.92%. The hesperidin content was 13.8% and total punicalagins as a sum of punicalagin A and B was 3.54%. Regarding the asiaticosides, the sum of asiaticoside, asiaticoside B and madecassoside on a dry basis was 8.1%.

### 3.2. EY Inhibited H_2_O_2_-Induced Proliferation Reduction in Human Dermal Fibroblasts

One hallmark of chronological skin aging is a slower skin cell proliferation rate [17]. Therefore, a functional assay to evaluate the ability of EY in promoting proliferation in aged human fibroblasts was evaluated.

Consistent with the reported effect of aging, our results show that NHDFs pre-treated with pro-oxidant (H_2_O_2_) conditioned medium displayed a significant decrease in cell proliferation rate (−9.2%) when compared to untreated NHDFs (Figure 2). These data supported the use of pro-oxidant conditioned medium in NHDFs as an adequate model to reproduce the reduced proliferation rate observed in aged skin fibroblasts.

When cells were supplemented with EY for 96 h, they recovered their proliferation capabilities. Additionally, a significant increase in cellular viability was observed at 0.001% and 0.0005%, with a 11.3% (*p* < 0.01) and 10.7% (*p* < 0.01) increase, respectively, raising their growth rate levels to those observed in non-aged fibroblasts. However, their viability was still lower than the control cells incubated with epidermal growth factor (EGF), which presented a 22.9% increase (*p* < 0.0001) (Figure 2).

### 3.3. EY Inhibited H_2_O_2_-ROS Production in Human Skin Fibroblasts

Free radicals and ROS have been reported to mediate the majority of the reactions leading to oxidative stress, which has been demonstrated to result in the cessation of DNA replication and eventually senescence in fibroblasts [50]. Additionally, ROS plays a major role in extrinsic skin aging, mainly caused by UVA radiation and pollution exposure, which promotes gene expression changes leading to collagen degradation and elastin accumulation, directly damaging extracellular matrix (ECM) collagen and inactivating MMPs inhibitors, responsible for protein degradation in the ECM [51]. Aged fibroblasts produce a greater amount of ROS, which further increases the expression of MMPs and inhibits TGF-β signaling, which leads to collagen fragmentation and decreased collagen biosynthesis, creating a positive feedback loop that accelerates dermal aging [2].

For these reasons, the goal of this assay was to assess the anti-aging effects of EY in human dermal fibroblasts, subjected to culture conditions emulating the aging process through quantification of the intracellular ROS levels. Thus, the effects on ROS production after treatment for 24 h with the tested sample were assessed in NHDFs. For this purpose, the conditioned medium with a senescence-inducing dose of H_2_O_2_ was applied for 3 h to NHDFs prior to exposure to the tested products.

As expected, 24 h after the dose of H_2_O_2_, NHDFs displayed a significantly higher level of intracellular ROS over untreated cells (+60.4%), which indicates that the treatment was useful to induce oxidative stress, a hallmark of aged skin fibroblasts. On the other hand, when cells were supplemented with EY for 24 h, ROS levels decreased. Treatment with the ingredient at 0.0005%, 0.005% and 0.01% decreased ROS levels by 16.9% (*p* > 0.05), 32.5% (*p* < 0.01), and 57.7% (*p* < 0.0001), respectively, compared to the untreated control (Control Aged), as shown in Figure 3.

### 3.4. Effect of EY on Telomere Length in Human Fibroblasts

The antiaging and protective capabilities of EY were evaluated after treatment in senescent normal human dermal fibroblasts (NHDFs) by quantifying telomere length. Telomere length quantification in each of the conditions was determined after inducing senescence with cell passages and doublings. Telomeres and SCR (reference housekeeping gene) were amplified using three technical replicates of gDNAs. To evaluate the correct amplification and specificity of primer pairs for the genes of interest, melting curves for each primer pair were calculated.

The results indicate that the human fibroblasts at 24 cell doublings without treatment significantly decreased telomere length by 0.42 ± 0.04 kb, compared to the normal control (not aged, T = 0). This confirmed that the cell model was indeed in senescence and reducing telomere length (data not shown).

When cells were treated with EY at 0.0001% throughout the entire culture period, telomere length displayed an increasing trend to prevent telomere shortening, with an average telomere length 0.21 ± 0.10 kb longer than the aged control. Conversely, the product at 0.0005% did not display any effect on telomere length versus the untreated control. Telomere shortening values were normalized versus the number of cell divisions underwent by cells in every condition. The number of cell doublings calculated through cell counting can be seen in Figure 4. In this case, the results show a statistically significant decrease in telomere shortening exerted by EY at 0.0001%. Under these conditions, telomeres were shortened at a rate of 0.010 ± 0.003 kb/division slower than untreated cells (*p* < 0.05) (Figure 4b). This implied a telomere shortening rate 57.7 ± 19.5% slower than the untreated aged control. No relevant effects were observed after treatment with the 0.0005% concentration.

Therefore, NHDFs treated with EY at 0.0001% showed an increase in their proliferative capacity compared to the control, at 29 cell doublings vs. 24 cell doublings after the fifth passage (Figure 4a).

### 3.5. Effect of EY on Telomere Length in Human Fibroblasts under Oxidative Stress Conditions

It is known that oxidative stress can lead to accelerated telomere length shortening, whereas antioxidants may delay attrition through their antioxidant activity [24,52].

Thus, we wanted to determine if EY, which is rich in phenolic antioxidant compounds, could also exert a telomere-protective effect under oxidative stress conditions. For this purpose, adult human dermal fibroblast cells were used to observe the effect of EY on telomeres under oxidative stress conditions, using 10 μg/mL H_2_O_2_, over a 6- and 8-week period. In this case, telomere length was measured using Life Length’s proprietary technology TAT^®^_._

TAT^®^ is a robust and reproducible high-throughput quantitative fluorescent in situ hybridization (HT Q-FISH) technology that allows for measuring telomere length in individual chromosomes in interphase cells and has significant advantages over other telomere testing methodologies. TAT^®^ technology was used because it allows us to determine not only the median telomere length in absolute units (base pairs bp) but also report the distribution and 20th percentile of telomere length, as well as inform ourselves of the percentage of critically short telomeres, <3 Kbp [43]. The median (50th percentile) is more representative compared to the mean because the telomere lengths are not distributed normally within the same sample. The 20th percentile is a second telomere length measurement obtained from the telomeric distribution and can be considered as a representative telomere length for the shorter telomeres, providing an additional point of comparison between samples. More specifically, it indicates the telomere length below which the lower 20% of the observed telomere lengths fall within a sample. On the other hand, the percentage of telomeres < 3 Kbp is an estimator of the critically short telomeres in the cells. Both of the latter variables (20th percentile length and percentage of telomeres < 3 kbp) are important because mounting scientific evidence shows that the short telomeres are responsible for causing cellular aging and its collateral effects [53]. This is due to critically short telomeres that increase the risk of cells entering in senescence, apoptosis and increased incidence of diseases [54], thus causing permanent and deleterious damage to the cells unless their attrition is lowered through a protective effect, or they are repaired by telomerase. Therefore, this study allows a more comprehensive analysis of the effect of EY on telomere dynamics. Representative TAT images obtained during this study can be seen in the Appendix A. Data for all variables determined are presented in Table 2 and Figure 5.

As can be seen, normal human dermal fibroblasts aged for 6 and 8 weeks and under oxidative stress with H_2_O_2_ showed decreased telomere length as well as a higher percentage of critically short telomeres compared to the unaged control. Conversely, all treated groups with EY (0.001%, 0.0005%, 0.0001%) resulted in a significantly longer median telomere length versus control both at week 6 and week 8. Additionally, all concentrations of EY resulted in a greater 20th percentile telomere length compared with the control and reduced the percentage of critically short telomeres (<3 kbp) both at week 6 and week 8, compared to the untreated control (Table 2).

Since cell replication is one of the main causes of telomere shortening and considering the prolonged period of cell expansion (8 weeks), the telomere length measurements performed were normalized by the population doubling levels (cell replication) for each condition and time point (Figure 5).

The telomere shortening rate was calculated using the following formula: Median telomere length (initial-final)/Population Doubling. After 6 weeks under oxidative cell culture conditions (10 μM-H_2_O_2_), the ingredient EY reduced the telomere shortening rate by 29–92%, being statistically significant for 0.0005% and 0.0001% of EY (*p* < 0.05, Figure 5). At week 8, the telomere shortening rate appeared to be lower in EY-treated groups, but these effects did not reach statistical significance. The result obtained in this study is consistent with the differences mentioned in the previous point and suggests a protective effect of EY on telomeres under oxidative stress conditions.

Contrary to the results obtained in previous studies, EY was not able to increase the proliferation of fibroblasts compared to the untreated control. In fact, at the highest concentration studied (0.001%), a decrease in cell proliferation was observed, mainly from week 6 (data not shown).

### 3.6. Effect of EY on Telomerase Activity in Skin Fibroblasts

Telomerase is an enzyme that can elongate and repair telomeres. Telomerase activity is found in some cells that divide continually and must maintain telomeres above a critical length to perform their functions. Thus, by promoting telomerase activity, it is possible to increase telomere length and consequently extend the number of cellular divisions that can take place without incurring chromosomal damage or telomere fusion. While this will not make cells immortal, it may extend their lifespan. To measure the effect of EY on telomerase activity, the quantitative polymerase chain reaction (PCR)-based telomeric repeat amplification protocol (Q-TRAP) was used. This method has the advantages of being a faster, more sensitive and higher-throughput format compared to the TRAP assay [55].

Thus, relative telomerase activity was measured in human fibroblasts over a 72 h period in the presence of different concentrations of EY. As can be seen in Figure 6, at 6 h, an increase in telomerase activity was observed in the cellular group treated with EY at 0.001%. The rest of the dosages presented a similar activity to that of the untreated control for the 6 h timepoint. However, after 24 h, the relative telomerase activity was 120–145% higher in EY-treated cells, at all dose levels, compared with control cells (*p* < 0.01). At 72 h, control and EY-treated groups were observed to possess similar levels of telomerase activity.

Previous studies have shown that telomerase activity is regulated in a cell-cycle-dependent manner, with telomere synthesis only occurring during the S phase in human cells [56]. Here, we observed that EY increases telomerase activity at 24 h, corresponding to a period during which cells start the exponential phase of their growth curve [57].

### 3.7. Antiglycation Assessment of EY

The glycation process, through the formation of advanced glycation end products (AGEs), has been recognized as one of the critical parameters that accelerate signs of skin aging, especially in skin exposed to environmental factors, such as ultraviolet radiation. AGEs alter skin physiology by impairing the deposition, organization, and physicochemical properties of dermal extracellular matrix components. Glycated fibers become rigid, lose elasticity and have reduced regenerative ability, leading to damage such as laxity, cracking, and thinning skin [58]. Counteracting glycation is considered an important anti-aging approach in the maintenance of healthy skin texture [35]. To determine whether EY could reduce or prevent the formation of AGEs, as a key reporter of skin aging processes, NHDFs were subjected to a 24 h treatment with the test product and exposed to UV radiation (as AGEs inducer). The generation of AGEs was quantified using the ELISA technique in cell lysates.

As it can be observed in Figure 7, upon exposure to UV light, AGE levels were increased by 56.9% (*p* < 0.001) with respect to non-irradiated cells, indicating that glycation was induced under the tested conditions. On the contrary, the tested sample at 0.01% significantly reduced the levels of AGEs by 81.4 ± 25.9% (*p* < 0.01) compared to UV-irradiated untreated cells, suggesting that the product might either reduce the production rate of AGEs or contribute to their clearance. A decreasing trend in AGE generation was also observed at 0.003% and 0.001% dose by 45% and 32%, respectively, although it did not reach statistical significance (*p* > 0.05) (Figure 7).

### 3.8. EY Inhibited Melanin Biosynthesis in Human Melanocytes

The pigmentary system is modified by the aging process, and uneven pigmentation is one of the major changes associated with aging [59]. Melanin, secreted by melanocytes, is the major pigment of human skin colour in the basal layer of the epidermis. Melanin is crucial for photo-protection from harmful ultraviolet (UV) sunlight damage [60]. However, various hyperpigmented skin disorders result from the overproduction and subsequent accumulation of melanin, such as freckles, age spots, melasma, and senile lentigo, which can be a distressing problem. This increased production is triggered by a variety of factors, but the main ones are sun exposure, age, hormonal influences, skin injuries and inflammation. UVA irradiation is suggested to contribute to melanogenesis by promoting cellular oxidative stress and impairing antioxidant defenses [61,62]. Thus, improving the capacity of antioxidant defenses to cope with oxidative insults is proposed to be beneficial in UVA radiation-induced melanogenesis [62].

The goal of this study is to assess the potential effect of EY, with proven antioxidant phenolic compounds, to inhibit melanin levels during UVA irradiation (mimic sun exposure) of normal human epidermal melanocytes (NHEM).

For the melanin quantification assay, cell viability assessment was performed in parallel and melanin levels were normalized to the number of living cells for each condition to avoid possible interferences due to the potential cytotoxicity induced by the UVA irradiation protocol. Melanin levels normalized to cell viability are shown in Figure 8a. The results indicate that the UVA irradiation protocol (5 J/cm^2^) significantly increased melanin levels by 40.8 ± 8.9% compared to the non-irradiated control. When EY was applied at 0.0005% and 0.005% concentrations for 7 days, the results show that treatments significantly decreased melanin levels by 40.0 ± 8.9% and 41.3 ± 8.9%, respectively (Figure 8a).

If we focus on cell survival analysis, the results from the MTT assay indicated that UVA irradiation significantly decreased cell viability by 24.9 ± 7.1% compared to the non-irradiated control. When EY was applied at 0.0005% and 0.005% concentrations, the results showed that the treatments significantly improved cell viability and reduced UVA damage by 19.9 ± 7.1% and 25.5 ± 7.1%, respectively (Figure 8b). This result proved that EY protected melanocytes from UVA-induced cytotoxicity.

## 4. Discussion

Some of the critical elements protecting skin cells from the inevitable aging process include cellular senescence, decreases in cellular DNA repair capacity and telomere shortening, oxidative stress, the mutation of extranuclear mitochondrial DNA, glycation, and chronic inflammation [3,63]. These factors seem to be at least partially caused or worsened by excessive reactive oxygen species (ROS) [13,64,65,66], and therefore treatments with antioxidants may be a plausible way to slow down or delay skin aging.

Many studies suggest that fruit and vegetable intake, or botanical-based dietary supplements, may prevent skin damage and cell aging by acting upon telomerase activity, telomere length and oxidative stress [22,67]. Ideally, the intake of a large variety of antioxidants seems to be more effective than a high concentration of a single antioxidant, suggesting complementary or synergistic effects [68,69,70,71,72]. Thus, dietary supplements containing botanicals and their constituents with proven antioxidant activity may be an adequate strategy to delay accelerated cell aging when exposed to pro-oxidant factors.

In this vein, the current study assesses the effects of a blend of four plant extracts: *Centella asiatica*, *Punica granatum* fruit, herba Cistanche and sweet orange extract. These extracts and their active compounds have been previously demonstrated to have an antioxidant effect, with several of them proven to modulate or reduce different cellular pathways of interest to reverse the skin aging process. The dried fleshy stem of the Cistanche genus (“Rou Cong Rong” in Chinese) has been used as a tonic in China for centuries. The chemical constituents of Herba Cistanche include phenylpropanoids glycosides (mainly verbascoside and equinacoside), iridoids, and polysaccharides. There has been an increasing number of studies focusing on its bioactivities, including antioxidation, neuroprotection, and antiaging [30,73,74]. Verbascoside, also known as acteoside, has been proven to have a wide range of activities, including antioxidant, anti-inflammatory, photoprotective, whitening and chelating effects [75,76,77]. In addition, verbascoside from Cistanche was found to increase telomerase activity in the heart and brain tissue in a mouse model of aging when it was given at a dose of 40 mg/kg for two weeks [78]. Echinacoside, the other phenylpropanoid present in the Cistanche extract, has also been proven to have several anti-aging related effects, including delaying cell senescence [79,80]. *Centella asiatica* (gotu kola) is a medicinal plant that has been used in folk medicine for hundreds of years as well as in scientifically oriented medicine [26]. The active compounds in *Centella asiatica* extracts include pentacyclic triterpenes, mainly asiaticoside, madecassoside, asiatic and madecassic acids, all of which are commonly known as asiaticosides. Centella extract has been commonly used as a dermo cosmetic botanical for many years, specifically to treat wounds, burns, psoriasis, and scleroderma. The mechanism of action involves promoting fibroblast proliferation and increasing the synthesis of collagen and intracellular fibronectin content [33]. Additionally, recent studies have addressed whether *C. asiatica* extracts may prevent or delay senescence [27].

Citrus fruits are a well-recognized source of bioactive compounds, with flavonoids being their major active compounds. The flavanone hesperidin is reported to be the most abundant flavonoid in oranges and lemons, with a wide range of biological and pharmacological properties, such as anti-inflammatory, antioxidant, lipid-lowering, and insulin-sensitizing properties [31,81]. Citrus fruits and hesperidin have also been shown to have anti-aging and photoprotective properties, by preventing the expression of matrix metalloproteinases (MMPs) in skin cells [32,82]. Hesperidin´s potential antiaging properties have been elucidated in yeast via the inhibition of ROS and UTH1 gene expression, as well as by increasing Sir2 activity [83]. Additionally, hesperidin exhibits protective effects against PM2.5-induced mitochondrial and DNA damage, cell cycle arrest, and cellular senescence through the ROS/JNK pathway [84].

Finally, the antioxidants of *Punica granatum* (pomegranate) fruit are widely known. Pomegranate is a rich source of polyphenols and several studies have shown the beneficial role of both pomegranate and its bioactives on reducing oxidative stress and lipid peroxidation through the direct neutralization of ROS, upregulating antioxidant enzymes, and modulating transcription factors such as nuclear factor κB (NF-κB) or peroxisome proliferator-activated receptor γ (PPARγ), among others [28,85,86]. Ellagic acid and punicalagin are the main bioactive compounds of pomegranate fruit extract that promote skin health by means of their antioxidant and anti-inflammatory effects or by inhibiting the tyrosinase enzyme [87,88,89]. Pomegranate extract also stimulates type I procollagen synthesis and inhibits MMP-1 (collagenase) production by dermal fibroblasts [34].

The purpose of this study was to determine the potential use of this multi-component plant extract (EY) to prevent skin aging. To this end, normal human dermal fibroblasts (NHDFs), the most important extracellular matrix-producing cell type in the dermis, were used and five key parameters where analyzed: cell proliferation, antioxidant activity, telomere length, telomerase expression and antiglycation properties.

ROS plays a major role in the skin aging of dermal fibroblasts, which promote gene expression changes leading to collagen degradation and elastin accumulation. Additionally, as we age, skin cells have lower regenerative and proliferative capabilities, and the accumulation of senescent cells is increased. Increased ROS levels also play a key role in cell senescence onset and maintenance. For these reasons, we examined the anti-aging effects of EY in stress-induced premature senescent fibroblasts through the quantification of cell proliferation and intracellular ROS levels. Stress-induced premature senescence (SIPS) fibroblasts were obtained by adding H_2_O_2_ [90,91]. SIPS shares some features with chronological or replicative senescence, including changes in the morphology of the cells, decreased cell proliferation and DNA synthesis, and increased β-galactosidase activity and oxidant generation [91,92]. Consistent with previous studies, our results also show that SIPS led to the loss of cell viability (Figure 2) and an increase in ROS production (Figure 3). However, EY treatment promoted an increase in fibroblast proliferation, suggesting that the compound possesses proliferative effects. In addition, EY possessed significant antioxidant effects in senescent fibroblasts, similar to those found in other studies with natural antioxidant compounds [91,93,94].

Replicative senescence is also strongly associated with telomere shortening [9,10]. Telomeres progressively shorten with age due to cumulative cell division cycles that are required for tissue repair and regeneration. However, this is not the only determinant for telomere shortening. Telomeres, due to the high guanine content in their sequence, are very sensitive to oxidative stress, alkylation and UV radiation, which are the leading causes of accelerated aging [24,95]. In fact, it is suggested that oxidative stress is an important modulator of telomere loss, and that telomere-driven replicative senescence is primarily a stress response. Therefore, delaying telomere shortening seems to be important in order to limit premature cell aging and senescence [13,96].

Telomeres in skin cells are particularly susceptible to accelerated shortening since skin cells have a high turnover, and the skin is highly exposed to external pro-oxidant factors such as solar radiation and pollution [14]. If telomere shortening is mediating the effects of ROS with regard to aging [14,52], antioxidants polyphenols present in EY may provide some protection against telomere shortening.

To study whether EY could reduce telomere shortening, two independent experiments were performed. In both assays, we proved that EY could partially reduce the telomere shortening rate in dermal fibroblasts (Figure 4b and Figure 5). First, telomere length quantification was determined in NHDFs cultured and grown during 24–30 cell doublings, inducing aging and telomere shortening due to cell passage, and treated with EY. Telomere length was measured using quantitative real-time polymerase chain reaction (qPCR). qPCR is the most widely used technique because of its relatively easiness to perform, and because it requires only a small amount of DNA [97]. The results of this study show that when fibroblasts were treated with EY at 0.0001%, during the entire culture period, the telomere shortening rate was 57.7% slower than untreated control (Figure 4b).

Based on the previous results on oxidative stress and accelerated telomere length shortening, we also determined whether the EY ingredient could also exert a telomere protective effect under oxidative stress conditions using H_2_O_2_. In this case, telomere length was analyzed using Life Length’s proprietary technology TAT^®^_._ Contrary to qPCR, this technique is more accurate and allows us to detect even subtle changes in telomeres through thousands of individual telomere measurements; however, it is much more technically demanding [43,98]. Furthermore, the TAT^®^ technology allows us to obtain information about the distribution of long and short telomeres and regarding the differences between individual cells and chromosomes that cannot be obtained with qPCR [43]. This information is an important point because the relative frequency of the shorter telomeres, which may trigger a cell cycle arrest, is a crucial factor that can induce senescence and affect cell viability and chromosome stability, and is highly associated with mortality [99,100]. The results of the present study suggest that EY may also have a protective effect on telomeres under oxidative conditions, resulting in higher median telomere and 20th percentile telomere length, as well as a lower percentage of short telomeres < 3 kbp (Table 2). Consistent with this observation, a lower telomere shortening rate was detected compared to the untreated control (Figure 5). Our results are similar to those of other studies with antioxidants using the same in vitro model and TAT^®^ analysis [44,101] and are in agreement with previous works demonstrating the protective effects of antioxidants on telomere length [24,52,70]. In this sense, EY may possibly delay the onset of replicative senescence by protecting the telomeres.

Besides the aforementioned results in telomere protection and reduced telomere shortening rate, EY also seemed to be capable of restoring telomerase activity (Figure 6). However, the role of telomerase, and especially its catalytic subunit (hTERT), in skin aging is not limited to protecting telomeres from shortening. hTERT has several regulatory properties in oxidative stress, inflammation, gene expression, and cell proliferation, as well as contributing to protecting the cells under stress conditions caused by external UVR and internal ROS [24].

Another important feature of aging and age-associated disorders is the gradual accumulation of non-enzymatic, covalent attachment of glucose molecules to proteins. This process, known as glycation, leads to the reversible production of reactive intermediates and ultimately to irreversible advanced glycation end products (AGEs). Proteins in the dermal matrix, such as collagen and elastin, and the cytoskeleton are particularly susceptible to glycation. Thus, glycated fibers become rigid, lose elasticity and have reduced regenerative capabilities, leading to skin laxity, cracking, and thinning [58]. The glycation process also causes deep changes in the behavior of dermal fibroblasts, reducing their proliferation and migration, while simultaneously disrupting collagen I maturation and preventing collagen deposition in the extracellular matrix [102]. In addition, AGEs induce cellular senescence, increasing the production of metalloproteinases (MMPs), the degradation of the extracellular matrix, and the ultimately the appearance of wrinkles, as well increasing melanin production, resulting in hyperpigmentation and uneven skin tone [58,103].

There are many studies investigating the potential substances that can cause the excessive accumulation of AGEs in tissues, and anti-AGE strategies has received high interest from pharmaceutical and cosmetic companies [25,35,104]. Diet control, reducing total calories and avoiding high-sugar foods may be effective in preventing the accumulation of AGEs in human skin [103]. However, as we age, these measures are insufficient. Additionally, topical products to treat glycation are scarce and generally inefficient, since the glycation process begins internally and moves outwards [105], and thus supplements containing glycation-preventing ingredients may be an alternative and complementary strategy to delay the skin aging process caused by AGEs. The accumulation of AGEs increases naturally with age but is enhanced by ultraviolet exposure, hyperglycemia and in chronic oxidative stress conditions [103]. Since oxidation is crucially involved in the formation of many AGEs, substances with antioxidative or metal chelating properties may also have antiglycation activity [64]. Thus, in the current study, the abundance of AGEs was evaluated in fibroblasts exposed to UV irradiation and cultured in the presence of EY. The results of our study show that EY significantly reduced the production of AGEs (Figure 7), suggesting that the product could be used to prevent or reduce glycation.

Finally, we examined whether this botanical blend could inhibit melanin production in normal human melanocytes exposed to UVA radiation. UVA radiation is suggested to contribute to melanogenesis by promoting cellular oxidative stress and impairing antioxidant defenses. The results of this study show a significant reduction in melanin production. This result thus opens the possibility that EY could be used to lessen hyperpigmentary disorders such as melasma, freckles, and age spots.

The mechanism by which this ingredient is capable of inhibiting melanin production is most likely due to the antioxidant activity of the phenolic compounds it contains. However, we cannot rule out its effect on the α-MSH-MC1R signal pathway, which is activated in UV-induced melanogenesis, since it has been shown that some of the active compounds present in EY have inhibitory effects on this pathway [74,76,84,106]. Additional studies would be needed to elucidate the exact mechanism.

While the results obtained in this work with EY are favorable, they are limited by the in vitro nature of the experiments. Additionally, the effects of the individual extracts used in the blend have not been assessed, although there is evidence from previous studies suggesting that they may contribute to delaying cell aging and telomere shortening. Furthermore, the activity of other antioxidant enzymes such as SODs, catalase, and glutathione-peroxidase/reductase has not been assessed. Future studies should examine whether these effects also occur in vivo. On the other hand, anti-aging clinical evaluation of this combination product has been completed, with findings to be published shortly.

## 5. Conclusions

In this research, we evaluated the anti-aging potential of a botanical blend comprising four standardized extracts (EY). To achieve these objectives, a series of preclinical studies have been performed. The results of these analyses prove that EY may counteract important mechanisms of skin aging. EY significantly reduced the telomere shortening rate in both stressed and non-stressed replicating human dermal fibroblasts. Additionally, it reduced the percentage of critically short telomeres (<3 kpb), which correlates with the risk of cells entering senescence. In addition, EY prevented the loss of proliferative potential and intracellular ROS formation on stress-induced premature senescence dermal fibroblasts, which may favor the synthesis of the structural proteins of the dermis, thus preserving its structural organization. Furthermore, EY, as an ingredient rich in antioxidants, significantly reduced the levels of AGEs in fibroblasts exposed to UV irradiation, suggesting that this product might either reduce the production rate of AGEs or contribute to their clearance. These results suggest that the combination of standardized extracts of pomegranate, sweet orange, Cistanche and *Centella asiatica* may have significant anti-aging effects and support the potential use of this combination as a new natural antiaging ingredient.

## Figures and Tables

**Figure 1 molecules-27-08101-f001:**
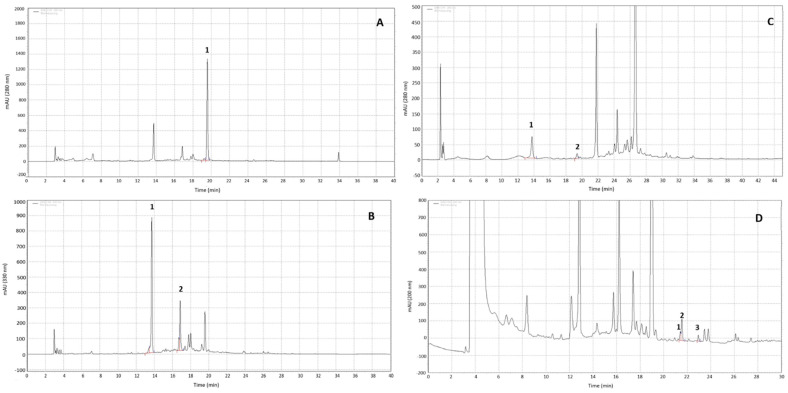
HPLC chromatograms of compound markers in EY. Chromatograms (**A**,**B**) are the result of applying the same method at different wavelengths ((**A**): 280 nm and (**B**): 320 nm) to quantify hesperidin (chromatogram (**A**), peak 1) and phenylpropanoids (chromatogram (**B**), peak 1 echinacoside, peak 2 verbascoside). Chromatogram (**C**) represent the base peak obtained by the method used to quantify punicalagins (peak 1, punicalagin (**A**), peak 2 punicalagin (**B**)) and Chromatogram (**D**) represent the base peak obtained by the method used to quantify asiaticosides (peak 1 asiticoside (**B**), 2 madecassoside, peak 3 asiaticoside). Details of the method used can be found in the Appendix A.

**Figure 2 molecules-27-08101-f002:**
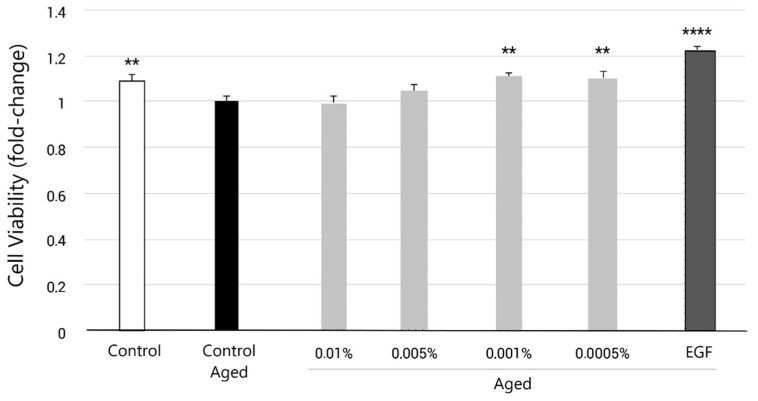
Bar graph presentation of fold change in the cell viability after pre-treatment of NHDFs with H_2_O_2_ (3 h) and subsequent treatment for 96 h with EY at 0.01%, 0.005%, 0.001% and 0.0005% concentrations (grey bars), compared to the untreated control (black bar, aged control). Epidermal growth factor (EGF) was used in parallel as a positive control (dark grey bar) and cell viability of NHDFs cultured in regular medium (non-aged control) was also measured (white bar). Data are expressed as the mean ± SEM. ** (*p* < 0.01) and **** (*p* < 0.0001) indicate significant differences compared to the aged untreated control (black bar).

**Figure 3 molecules-27-08101-f003:**
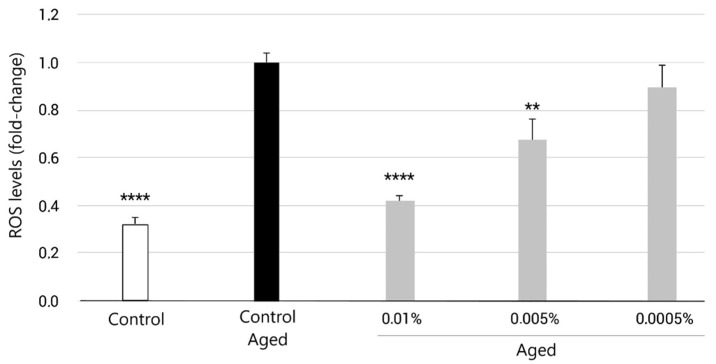
Bar graph representing fold change in intracellular relative ROS levels after 3 h pretreatment with H_2_O_2_ conditioned medium, and subsequent treatment for 24 h with EY at 0.01%, 0.005%, 0.0005% concentrations (grey bars), compared to the untreated control (black bar, aged control). Intracellular ROS levels of NHDFs cultured in regular medium (non-aged control) was also measured (light grey bar). Data are expressed as the mean ± SEM. ** (*p* < 0.01) and **** (*p* < 0.0001) indicate significant differences compared to the aged control.

**Figure 4 molecules-27-08101-f004:**
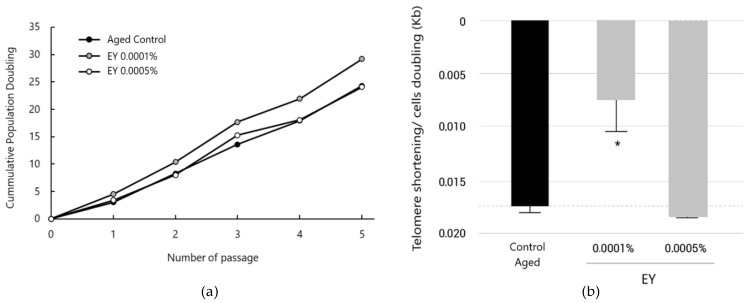
(**a**) Growth curves of Normal Human Dermal Fibroblasts treated with EY at 0.0005% and 0.0001% compared to the untreated control (black circle). Each point on the population curve represents the average of the triplicates for each cell passage. (**b**) Bar graph showing aged-induced telomere shortening per cell division in normal human dermal fibroblasts and treatment with EY at 0.0001% or 0.0005%, compared to aged control. Data are expressed as the mean ± SEM. * (*p* < 0.05) indicate significant differences compared to Aged untreated control (black bar).

**Figure 5 molecules-27-08101-f005:**
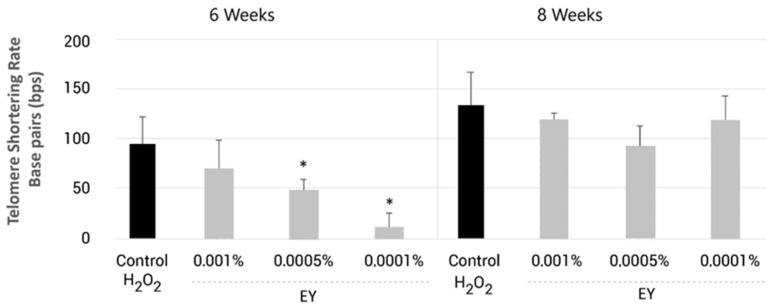
Bar graph showing aging plus oxidation-induced telomere shortening per cell division (telomere shortening rates of the median telomere length) in normal human dermal fibroblasts treated with EY at 0.0001%, 0.0005% or 0.001% (grey bars) and compared to oxidated untreated control (black bars) after 6 and 8 weeks of incubation. Data are expressed as the mean ± SD. * (*p* < 0.05) indicate significant differences compared to untreated control.

**Figure 6 molecules-27-08101-f006:**
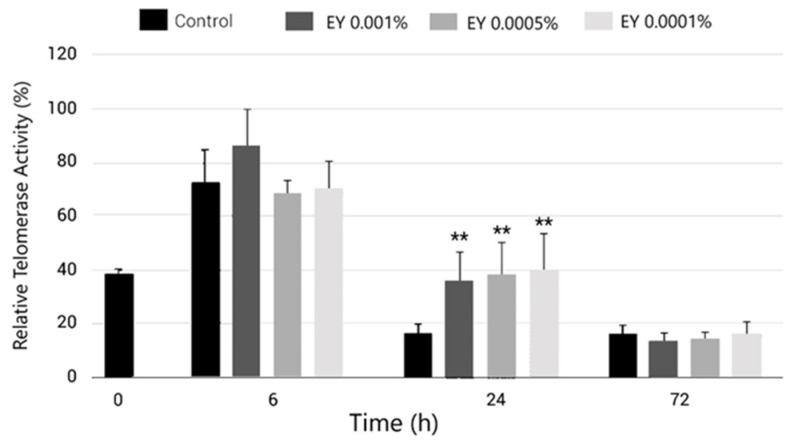
Bar graph showing relative telomerase activity in normal human dermal fibroblasts treated with EY at 0.0001%, 0.0005% or 0.001% (grey bars) and compared to oxidated untreated control (black bars) after 6, 24 and 72 h of incubation. Data are expressed as the mean ± SD. ** (*p* < 0.01) indicate significant differences compared to untreated control.

**Figure 7 molecules-27-08101-f007:**
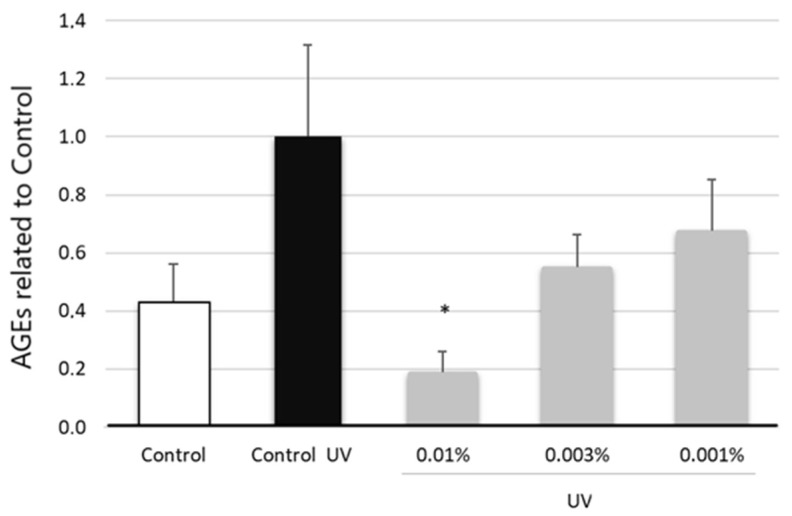
Bar graph representing AGE levels relative to total protein content and normalized versus untreated UVB-irradiated control (black bar) in NHDF lysates treated with EY at 0.01, 0.003 and 0.005% (grey bars) for 24 h. AGEs levels of non-irradiated NHDFs were also measured (control, white bar). Data are expressed as the mean ± SEM. * (*p* < 0.05) indicate significant differences compared to the UVB-irradiated control.

**Figure 8 molecules-27-08101-f008:**
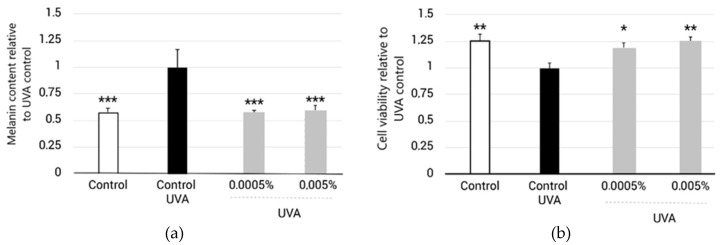
(**a**) Bar graph representing melanin levels after treatment with EY for 7 days at 0.0005% and 0.005% in NHEM submitted to the UVA irradiation protocol and normalized to Control + UVA (black bar). Melanin content of non-irradiated NHEM was also measured (white bar). (**b**) Bar graph representing cell viability levels after treatment with EY for 7 days at 0.0005% and 0.005% in NHEM submitted to the UVA irradiation protocol and normalized to Control + UVA (black bar). Cell viability of non-irradiated NHEM was also measured (white bar). In (**a**,**b**), data are expressed as the mean ± SEM. * (*p* < 0.05) ** (*p* < 0.01) and *** (*p* < 0.001) indicate significant differences compared to the UVA-untreated control (black bar).

**Table 1 molecules-27-08101-t001:** Identified compounds by HPLC, their retention time, wavelength and method used for their quantification.

Peak	Compound	RT (min)	Wavelength (nm)	Method
A 1	Hesperidin	19.5	280	1
B 1	Echinacoside	13.7	330	1
B 2	Verbascoside	16.8	330	1
C 1	Punicalagin a	13.7	280	2
C 2	Punicalagin b	19.3	280	2
D 1	Asiaticoside B	21.2	200	3
D 2	Madecassoside	21.5	200	3
D3	Asiaticoside	23	200	3

**Table 2 molecules-27-08101-t002:** Effect of EY on telomere length under oxidative stress conditions. Median telomere length (50th percentile), 20th percentile telomere length, percentage of short telomeres (<3 kbp) and the percentage of variation from the control ((Δ(Treated − Control)/Control) × 100) H_2_O_2_.

		Median Length (bp)	Variation from Untreated Control (%)	20th Percentile Length (bp)	Variation from Untreated Control (%)	Telomeres <3 kbp (%)	Variation from Untreated Control (%)
Start (Week 0)	Unaged Control	9955		6704		3.9	
Week 6	Control H_2_O_2_	8292		5356		5.8	
	EY 0.001%	8854	6.78	5819	8.64	4.6	−20.69
	EY 0.0005%	9356	12.83	6385	19.21	3.6	−37.93
	EY 0.0001%	9523	14.85	6530	21.92	3.5	−39.66
Week 8	Control H_2_O_2_	6813		3621		16.4	
	EY 0.001%	7088	4.04	3765	3.98	16	−2.44
	EY 0.0005%	8013	17.61	4627	27.78	11.6	−29.27
	EY 0.0001%	7626	11.93	4718	30.30	9	−45.12

## Data Availability

Not applicable.

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
