# Peer review of "In Vitro Determination of the Skin Anti-Aging Potential of Four-Component Plant-Based Ingredient"

_molecules, 2022, doi:10.3390/molecules27228101_

Round 1

Reviewer 1 Report

Very interesting manuscript, that summarized numerous methodological approaches for in vitro anti-aging effects of 4 botanical extracts.

I am curious about the sample preparation for cell culture experiments. Could the authors state the final DMSO concentration that was applied to the cells?

Author Response

Comments and Suggestions for Authors

 I am curious about the sample preparation for cell culture experiments. Could the authors state the final DMSO concentration that was applied to the cells? 

We sincerely thank the reviewer for his/her time and comments. Regarding your question, the final concentration of DMSO in the cells culture media was maximum 0.1% at which no detrimental effect on cell growth or toxicity was detected. This information has been included in the Materials and Method Section.

Reviewer 2 Report

The article is well designed and written, minor revisions are suggested:

- The introduction is clear, but the authors should describe more about what there is in the literature on the botanical compounds chosen for the treatment.

Methods:

- Add a topic about the cells used and their maintenance

- line 142- describe all dilutions used

Results

- In figure 1, the author must separate the table from the chromatograms.Even if the information is complementary.

With the results, it is possible to observe the reduction of peroxide. However, this is just one of the ROS involved in skin aging. Authors should assess the activity of antioxidant enzymes such as SOD, CAT, GPX, etc. Also, which other species are diminished with treatment?

Does the TAT provide, in addition to quantifications, images? It would be quite interesting to have fluorescent images representative of the graphics.

Discussion:

The discussion is sometimes long and repetitive. Write in a more direct and compact way, without repeating the results of the work.

Author Response

Comments and Suggestions for Authors

We sincerely thank the reviewer for his/her time and the revision suggested.

- The introduction is clear, but the authors should describe more about what there is in the literature on the botanical compounds chosen for the treatment.

We have included in the introduction more details on the botanical compounds chosen for the treatment.  Although we did not want to extend in excess since in the Discussion section more detailed information is also given on each of the extracts that are part of the formula.

Methods:

- Add a topic about the cells used and their maintenance.

We have included a new section in the materials and methods with a brief description of the cells used and their maintenance.

- line 142- describe all dilutions used.

We have not detailed all the dilutions used in the studies since we think that they do not provide relevant information and would be excessive. However, we have described in more detail how the working solutions is prepared and how we make the dilutions. Hope this information satisfy your request.

Results

- In figure 1, the author must separate the table from the chromatograms. Even if the information is complementary.

We have separated the Figure 1 in Table 1 and Figure 1

- With the results, it is possible to observe the reduction of peroxide. However, this is just one of the ROS involved in skin aging. Authors should assess the activity of antioxidant enzymes such as SOD, CAT, GPX, etc. Also, which other species are diminished with treatment?

The Fluorometric Intracellular ROS Assay Kit used in our study is from Sigma. And according to their Technical Sheet, this Kit provides a sensitive, one-step fluorometric assay to detect total intracellular ROS, but especially superoxide and hydroxyl radicals.

(Ref:https://www.sigmaaldrich.com/deepweb/assets/sigmaaldrich/product/documents/641/780/mak144bul.pdf).

We agree that it would be interesting to further explore antioxidant activity and study whether the tested ingredient could affect the activity of key antioxidant enzymes like glutathione peroxidase (GPx) and superoxide dismutase (SOD) and catalase (CAT). So, we will include this comment in the discussion section.

Does the TAT provide, in addition to quantifications, images? It would be quite interesting to have fluorescent images representative of the graphics.

We are including in the Supplementary section only 4 representative TAT images. One at Week 0, one at week 6 plus Oxidation, and one at week 6 with oxidation and with 0.0001% of the tested sample. 

At the same time, we would like to clarify that the full set of images for the entire study would be too excessive and laborious to provide and include in the manuscript. Kindly note that each sample is tested in quintuplicate using 5 separate wells and for each well, 10 images are approximately obtained by the microscope. So, we are analysing hundreds of photos for the entire study which is quite excessive to include in the manuscript.  

Additionally, please note that the analysis of the images and the measurement of the fluorescence intensity from the images is done automatically by the microscope that has the capacity to detect the smallest differences in fluorescence intensity that the eye cannot appreciated. These fluorescence intensities are then translated to telomere length (in base pairs) by our software automatically using a standard curve build by 7 control cell lines. The analysis is performed in 3Z (3 planes) to achieve higher precision. This means that it is not humanly possible to "see" the telomere length measurements are generated by analyzing of hundreds of thousands of telomeres in each sample. So, although some images can be shared it would not be possible to visualize the numerical result which is the median of hundreds of thousands of measurements per sample.  

Finally please note that the TAT technology has a CLIA certification due to its robustness and precision and the protocol followed for the measurements is described in the publication of the validation of the technology that is mentioned and included in the manuscript (https://pubmed.ncbi.nlm.nih.gov/31956299/) in case further information is required.

 Discussion:

The discussion is sometimes long and repetitive. Write in a more direct and compact way, without repeating the results of the work.

We have modified the discussion and have eliminated some redundant information. I hope now you find it less repetitive.